# Aprotinin (II): Inhalational Administration for the Treatment of COVID-19 and Other Viral Conditions

**DOI:** 10.3390/ijms25137209

**Published:** 2024-06-29

**Authors:** Juan-Fernando Padín, José Manuel Pérez-Ortiz, Francisco Javier Redondo-Calvo

**Affiliations:** 1Department of Medical Sciences, School of Medicine at Ciudad Real, University of Castilla-La Mancha, 13971 Ciudad Real, Spain; fernando.padin@uclm.es; 2Facultad HM de Ciencias de la Salud, Universidad Camilo José Cela, 28692 Madrid, Spain; josemanuel.perez@ucjc.edu; 3Instituto de Investigación Sanitaria HM Hospitales, 28015 Madrid, Spain; 4Department of Anaesthesiology and Critical Care Medicine, University General Hospital, 13005 Ciudad Real, Spain; 5Translational Research Unit, University General Hospital and Research Institute of Castilla-La Mancha (IDISCAM), 13005 Ciudad Real, Spain

**Keywords:** proteases, aprotinin, COVID-19, kinin–kallikrein system (KKS), renin–angiotensin–aldosterone system (RAAS), angiotensin-converting enzyme type 2 (ACE2), antifibrinolytic, inhalational administration, pharmacodynamic, pharmacokinetic

## Abstract

Aprotinin is a broad-spectrum inhibitor of human proteases that has been approved for the treatment of bleeding in single coronary artery bypass surgery because of its potent antifibrinolytic actions. Following the outbreak of the COVID-19 pandemic, there was an urgent need to find new antiviral drugs. Aprotinin is a good candidate for therapeutic repositioning as a broad-spectrum antiviral drug and for treating the symptomatic processes that characterise viral respiratory diseases, including COVID-19. This is due to its strong pharmacological ability to inhibit a plethora of host proteases used by respiratory viruses in their infective mechanisms. The proteases allow the cleavage and conformational change of proteins that make up their viral capsid, and thus enable them to anchor themselves by recognition of their target in the epithelial cell. In addition, the activation of these proteases initiates the inflammatory process that triggers the infection. The attraction of the drug is not only its pharmacodynamic characteristics but also the possibility of administration by the inhalation route, avoiding unwanted systemic effects. This, together with the low cost of treatment (≈2 Euro/dose), makes it a good candidate to reach countries with lower economic means. In this article, we will discuss the pharmacodynamic, pharmacokinetic, and toxicological characteristics of aprotinin administered by the inhalation route; analyse the main advances in our knowledge of this medication; and the future directions that should be taken in research in order to reposition this medication in therapeutics.

## 1. Introduction

The most common respiratory tract infections in humans are caused by viruses including adenoviruses, myxoviruses, orthomyxoviruses, rhinoviruses, and respiratory syncytial virus hantaviruses [1]. In addition, many of these common infections, such as influenza viruses or coronaviruses, share the same two-step mechanism of entry into the respiratory tract cell, by the cleavage and activation of viral capsid anchoring proteins by the proteases present in human host cells [2,3]. Based on this mechanism of infectivity, aprotinin has been proposed as an antiviral drug that also helps to alleviate symptomatic inflammatory processes associated with the infectious process [4]. This is due to its ability to re-establish the protease–antiprotease balance of respiratory function that is disturbed following infection. In this review, we will discuss the most relevant pharmacodynamic, pharmacokinetic, and toxicological aspects of the antiviral action of aprotinin when administered by inhalation.

## 2. Development of Aprotinin for Respiratory Viral Infections

Aprotinin was discovered in 1930 and is an effective panprotease inhibitor: a “magic bullet”, as Paul Ehrlich would say. It was initially used intravenously as an antithrombotic and anti-inflammatory medication in cardiac and non-cardiac surgery to reduce bleeding and limit the need for blood transfusions [5] (Figure 1A). However, due to the misinterpretation of adverse events in clinical trials and controversy in the literature, the use of aprotinin was almost discontinued for a decade worldwide. Between 2015 and 2020, after further re-evaluation of the safety data in these clinical trials, the restrictions on use were lifted by drug agencies [6]. Despite this, aprotinin’s unique characteristics as a highly stable polybasic protein with a broad spectrum to inhibit enzymes with serine-protease activity with great capacity meant that it had no shortage of potential repositioning options in other areas of therapeutics. From the early 1980s, the virologist Zhirnov and his collaborators studied aprotinin as an antiviral drug candidate to treat infections by respiratory viruses that infect cells using host proteases, focusing mainly on influenza virus [7,8], and on the treatment of bronchopneumonia caused by viruses in the Orthomyxoviridae and Paramyxoviridae families [9]. This led the research group to conduct a clinical trial using aerosolised aprotinin to treat infections with these viruses [10], and to market in Russia a drug called Aerus^®^ for the treatment of seasonal influenza [4] (see Figure 1B). Other in vitro and in vivo studies in experimental animals soon began to emerge, demonstrating the activity of aprotinin against other viruses such as Alphavirus [8], Sendai [11], rotavirus [12], myxovirus [13], herpesvirus [14,15], Dengue [16], and West Nile [16,17], among others. 

With the advent of the COVID-19 pandemic, the need arose to identify antiviral drugs that showed activity against Severe Acute Respiratory Syndrome (SARS) caused by coronavirus type 2 (CoV-2; SARS-CoV-2). Aprotinin was well positioned for this, given that the virus uses host proteases in its viral infection mechanism. Bojkova (2020) and Bestle (2020) with their collaborators were the first to demonstrate in vitro that aprotinin can inhibit the replication of SARS-CoV-2 [18,19]. Our research group, in a phase III clinical trial called “Aprotinin Treatment Against COVID-19” (ATAC), then evaluated the efficacy and safety of aprotinin and demonstrated that when administered by the inhalation route, it was able to reduce oxygen requirements and hospital admission time [20] and decrease the viral load in patients admitted with a moderate to severe prognosis [21].

## 3. Pharmacodynamic Actions of Aprotinin

SARS-CoV-2 uses proteases released into the extracellular space, or present in the plasma membranes of host epithelial cells, to cleave the spicule protein (S) from the viral capsid, and thus infect the cell, through anchoring to targets such as angiotensin-converting enzyme type 2 (ACE2). In this infectious process, the virus will provoke an imbalance by increasing protease activity relative to antiprotease activity [22,23]. This is of great importance in the regulation of lung and immune function in the respiratory tract [24]. This imbalance explains much of the COVID-19 disease (Figure 2).

Aprotinin is a broad-spectrum inhibitor of serine protease-type enzymes present in the digestive tract and lung (e.g., transmembrane serine protease 2 [TMPRSS2], trypsin, and chemotrypsin); in the blood participating in coagulation (e.g., plasmin, FXII, activated protein C, thrombin); and involved in innate immunity (e.g., neutrophil elastase, plasma and tissue kallikrein, and complement factors) [5,25,26]. Inhibition of the physiological activity of these proteases explains their pharmacological actions (see Table 1). 

### 3.1. Antiviral Activity

The antiviral activity of aprotinin is due to its non-specific inhibition of host proteases that are required for the proteolytic cleavage of the viral protein S, which is involved in the infectious process of entry through ACE2. In addition, during this entry process, the virus stimulates the increased synthesis, release, and action of many proteases, which will also contribute to enhancing its infective capacity and participate in the inflammatory process. For this reason, selective serine-protease TMPRSS2 inhibitor antiviral drugs have been proposed for the treatment of COVID-19 [57]. However, because they do not target the entire spectrum involved in the infective process of SARS-CoV-2 (e.g., cathepsins, trypsin, fibrinogen, or kallikreins), they have reduced efficacy and do not prevent many aspects of the disease. For this reason, broad-spectrum inhibitors such as aprotinin or α1-antitrypsin have shown greater efficacy than more selective drugs [20,21,58] (Figure 2). 

Cathepsins are proteases with great relevance in SARS-CoV-2 infection. They are not only secreted to the cell exterior, thereby favouring the anchoring of the virus to ACE2 to allow its entry through an endocytic pathway, but they also participate in the lysosomal maturation of viral proteins and the release and spread of viral progeny [59,60]. Aprotinin can interfere with cathepsin function [29,32]. Thus, like α1-antitrypsin [58], it can prevent entry via the endocytic pathway and the cell-to-cell passage of viruses. In viruses such as human immunodeficiency virus type 1 (HIV-1), other proteases such as T-lymphocyte tryptase type 2 (TL2) can also participate in these processes of entry, syncytium formation, and viral replication in CD4+ cells, where aprotinin also shows an inhibitory capacity [34,35]. Furthermore, cathepsins, among other locations, are also present in cellular organelles such as lysosomes and thus influence the trafficking of SARS-CoV-2 specific proteins such as the accessory protein Open Frame Reading 3a (OFR3a), which is of great importance in virus infectivity and new virion formation [61,62]. Therefore, when aprotinin inhibits cathepsins, it causes these viral proteins and new virions to be redirected towards a degradation pathway in multivesicular bodies [63]. Moreover, cathepsin L is involved in the upregulation and processing of heparinase, resulting in the release of viral progeny and their propagation [64,65].

Coronaviruses, such as Middle East Respiratory Syndrome (MERS)-CoV, are generally known to recognise the enzyme dipeptidyl peptidase-4 (DPP4) to anchor to the host cell membrane in their infective mechanism [66]. The use of proteases as anchoring proteins, such as ACE2 or DPP4, is common in these viruses [67]. SARS-CoV-2 has been suggested to use DPP4 as a co-receptor or auxiliary protein of ACE2 in the viral entry process [68,69]. This is supported by the fact that ACE2 is co-expressed with DPP4 in many human tissues, and because both show high levels in lung alveoli [69,70]. Moreover, there is functional cross-talk between both peptidases, where their activity is regulated reciprocally and in the same direction [71]. Indeed, DPP4 inhibitory drugs such as gliptins have been associated with reduced viral entry, replication capacity, and avoidance of the cytokine storm that occurs in COVID-19, especially in diabetic patients [72,73]. One of the advantages of aprotinin is that it also has the ability to inhibit DPP3 and DPP4 [36,37], another mechanism contributing to its antiviral activity. 

In summary, due to the ability to inhibit multiple proteases (for example, cathepsins), aprotinin has antiviral actions by preventing the viral attachment to the target protein (by preventing activation of viral protein S); penetration (prevent endocytosis or syncytium formation); the replication, maturation, and trafficking of viral proteins to lysosomes; and the assembly of virions and their release (for example, by inhibiting heparanase or angiotensinase C). The proteases it inhibits and their involvement in the viral infection mechanism are summarised in Table 1.

### 3.2. Anti-Inflammatory Activity

The inflammatory process in COVID-19 cannot be understood without understanding the close relationships between the kinin–kallikrein system (KKS), the renin–angiotensin–aldosterone system (RAAS), and the complement system [74,75]. On the one hand, the infectious process of SARS-CoV-2 induces a downregulation of ACE2 [76,77,78] that triggers an imbalance between the three systems, which mutually autoregulate one another [75]. This results in the increased production of angiotensin II [79,80] and des-Arg9-bradykinin, two mediators with a high inflammatory capacity [81,82]. However, there is an imbalance between the action of proteases and antiproteases. Finally, KKS is part of the contact system of innate immunity (coagulation factor XII, high molecular weight kininogen [HMWK], and pre-kallikreins), which activates complement and causes neutrophilia [82,83,84]. The sum of these three imbalances forms the basis of the inflammatory process in COVID-19. 

The anti-inflammatory action of aprotinin is multifaceted and is based on re-establishing the imbalance between (i) the KKS and RAAS, (ii) the action of proteases and antiproteases, and (iii) the contact system and complement activity of innate immunity. We will discuss how aprotinin regulates these imbalances, rather than the inflammatory actions of mediators such as bradykinin or angiotensin II, which have already been adequately explained in the following manuscript: Aprotinin (I). Understanding COVID-19 disease to understand its mechanism of action.

#### 3.2.1. Aprotinin and Its Ability to Re-Establish the Imbalance between KKS and RAAS in COVID-19

In COVID-19 disease, there are high levels of bradykinins due to an increase in their synthesis [85] and an alteration in their degradation due to an elevated ACE/ACE2 ratio [81]. Increased bradykinin levels and the downregulation of (membrane) ACE2 are also associated with elevated angiotensin II levels [86,87], as these enzymes show a greater affinity for bradykinin degradation than for angiotensin II itself [84,88,89]. There is therefore a saturation in its enzymatic activity. Aprotinin is a potent inhibitor of plasma and tissue kallikreins [27], which are the main source of bradykinin production [90]. Therefore, a decrease in their production will contribute to the restoration of the imbalance. However, there are other sources of angiotensin II production beyond ACE itself: endopeptidases such as the chymase released by mast cells [91], cathepsin G, chemostatin, and even tissue kallikreins themselves also produce angiotensin II [92,93,94,95]. These proteases are inhibited by aprotinin because they are serine proteases [94] (see Table 1 and Figure 2). This inhibitory capacity has been improved with the synthesis of different recombinant aprotinin mutations that have a high inhibitory capacity for these enzymes [29,31].

Moreover, aprotinin is not only involved in re-establishing the KKS–RAAS balance by affecting kallikreins and thus bradykinin. It also inhibits DPP3 and 4 [36,37]. Angiotensin II, as found in renal tubular cells, is known to cause an increase in DPP4 activity [96,97]. The enzyme, through its interaction with proteins such as adenosine deaminase, participates in T cell, B cell, and myeloid cell (dendritic cells and macrophages) activation processes, cell proliferation, and the transendothelial migration of inflammatory cells [98]. Angiotensin II and DPP4 thus contribute to the inflammatory process. The importance of this enzyme is that it may participate in the cytokine response of COVID-19 and explain the differential involvement of diabetic or obese patients in the disease [69]. By inhibiting DPP3 and 4 [36,37], aprotinin prevents the inflammatory cascade triggered by angiotensin II.

#### 3.2.2. Aprotinin Restores Protease and Antiprotease Balance in the Lung

The binding of SARS-CoV-2 to ACE2 is a process that is intrinsic to the upregulation of the metalloprotease adammalysin-17 (ADAM17) [99] and, consequently, the cleavage of ACE2 from the plasma membrane into its soluble form initiates the inflammatory process by two main steps: on the one hand, the release of inflammatory mediators such as tumour necrosis factor alpha (TNFα) or interleukin (IL)-6 in response to ADAM17 activity [100,101]; and on the other hand, the increase in angiotensin II and bradykinins results from the imbalance in the activation of RAAS and KKS or innate immunity contact system, which is associated with the viral entry mechanism itself. This results in the exacerbated release of proteases (e.g., neutrophil elastase, cathepsins, trypsin, kallikreins, and chymase) from inflammatory cells (e.g., mast cells or neutrophils), or from the epithelial cells themselves (e.g., exocrine glands or endothelial cells) [102,103]. The overactivation of proteases leads to damage of the extracellular matrix, which can form exosomes that impede the action of antiproteases [104]. In addition, proteases such as matrix metalloprotease-12 (MMP-12) or neutrophil elastase, whose increased release is linked to worse prognosis, degrade antiproteases such as α1-antitrypsin to form C-terminal peptides [105,106]. Proteases and neutrophil-induced oxidative stress itself transform antiproteases by glycosylation and sialylation to increase their anti-inflammatory properties [107]. However, these changes have no impact on their activity nor do they improve prognosis [106]. Consequently, there is an imbalance between the action of proteases and antiproteases. These imbalances can be caused by the action of viruses affecting the pulmonary tract [108,109,110,111] and are particularly affected in patients with COVID-19, where much of the pathology can be explained by a dysregulation in the activation of the contact system of the innate immune system [112,113,114] (Figure 2). Aprotinin can restore these imbalances. The inhibition of multiple proteases such as kallikrein, plasmin, thrombin, cathepsins, trypsin, chymase, chymostatin, or neutrophil elastase [4,5,26] not only prevents S-protein cleavage and thus the entry of SARS-CoV-2 into the host cell but also inhibits the activation of associated inflammatory processes and the release of cytokines such as IL-6 or TNF-α as a consequence of these events [115]. This prevents the transmigration of inflammatory cells such as monocytes and granulocytes [4,116,117] and the further release of proteases and the contents of their azurophilic granules [118,119]. Aprotinin has not only been shown to inhibit proteases such as neutrophil elastase but it has also been shown in numerous studies to inhibit their release from neutrophils under various pathological and experimental conditions [25,120]. Furthermore, in inflammatory cells such as neutrophils, one of their most important actions is their ability to inhibit enzyme-induced nitric oxide synthase (iNOS) and thus the generation of free radicals [52,53,54]. By inhibiting this, it prevents the transformation and cleavage of natural proteases such as α1-antitrypsin and the re-establishment of imbalances in the lung (Figure 2).

#### 3.2.3. Aprotinin Re-Establishes the Contact System and Complement Activity of the Innate Immune System

SARS-CoV-2 possesses structural proteins that can activate the blood complement pathway per se. For example, the N protein enhances the activation of the complement lectin pathway [121], and the S protein activates the alternative complement pathway by binding to heparan sulphate on cell surfaces [122] and to C4a [123]. Both complement pathways converge in the activation of the C3a and C5a anaphylatoxins [124,125]. Moreover, through the process of viral entry into the host cell, bradykinins, kallikreins, HMWK, plasmin, and coagulation factor XII are formed and play an active part of the contact system of innate immunity [126]. In particular, plasmin and coagulation factor XII are involved in the activation of the classical complement pathway [126,127] (Figure 3).

In patients undergoing myocardial revascularisation, aprotinin partially inhibits the activity of the classical complement pathway in a dose-dependent manner [128]. The multifunctional receptor gC1q, which is involved in this classical complement pathway, can be activated by HMWK serine proteases and coagulation factor XII [129] and this occurs even in the absence of immunoglobulins [130]. Aprotinin being a potent inhibitor of these proteases [4,5,26], its ability to inhibit this classical pathway is not surprising. Moreover, in patients undergoing haemodialysis or extracorporeal circulation, aprotinin also completely inhibits the alternative complement pathway [25,131]. Finally, the lectin pathway of the complement system operates through the activation of a mannose-associated serine protease (MASP). Aprotinin inhibits MASP types 2 (MASP-2) and 3 (MASP-3) and is therefore also a potent inhibitor of this lectin pathway [47,48,49] (Figure 3 and Table 1). 

### 3.3. The Thromboinflammatory Activity of Aprotinin

Dr Rudolf Virchow established the three factors that contribute to the formation of a vascular thrombus: (a) damage to the vascular endothelium; (b) states of hypercoagulability; and (c) impairment of vascular blood flow. All three elements are involved in the embolisms caused by SARS-CoV-2. 

First, endothelial damage occurs as SARS-CoV-2 infects endothelial cells [132] as well as vascular basement membrane pericytes [133,134], resulting in endothelial dysfunction [135]. Within the infectious process, the release of proteases by the endothelial cell contributes to its own damage. For example, the release of heparanase (a process involving cathepsins or fibrinogen) promotes damage to the endothelial glycocalyx [136] and the release of inflammatory factors [137]. Aprotinin prevents SARS-CoV-2 from infecting vascular cells, thus preventing the cause of damage [18]. Moreover, it inhibits the degradation of the extracellular matrix by preventing the degradation of heparan sulphate via heparinase [138].

Aprotinin can also prevent other vascular damage factors, such as by inhibition of the production of reactive oxygen species (ROS) by increasing the expression of haem oxygenase-1 [139]; inhibition of iNOS [52,54]; and inhibition of the activation of the thrombin protease-activated receptor (PAR) types 1 and 2 (PAR-1 and PAR-2), whose activation is implicated in tissue remodelling and fibrosis [42,43,44,45]. Finally, it can inhibit processes of platelet adhesion to damaged endothelium by mechanisms that are independent of nitric oxide generation [140]. Secondly, there is a state of hypercoagulation. The pulmonary microvascular thromboses that occur in COVID-19 disease occur largely because of the relationship between the activation of the innate immune system (contact system and complement) and coagulation. This particular type of pulmonary embolism is known as “thromboinflammation” and has a different pathophysiological basis than other vascular embolisms [141]. This is because the presence of viral antigens leads to complement activation [121,122,123], and the viral infection process itself contributes to the release of mediators such as bradykinins, kallikreins, HMWK, plasmin, and coagulation factor XII, which form an active part of the contact system of innate immunity [126]. Both activation of the complement system [134,142] and the contact system [143] contribute to the activation of coagulation pathways in SARS-CoV-2 infection. The contact and complement systems ultimately activate neutrophils by the secretion of proteases such as neutrophil elastase from their azurophil granules, which contribute to the infectious process, inflammation, endothelial damage, formation of Neutrophil Extracellular Traps (NETs) and aggravate, through a vicious cycle, the thromboinflammatory state [144,145]. Indeed, NET formation itself initiates the extrinsic coagulation and contact pathways by releasing tissue factor, thereby triggering factor XII activation and trapping and activating platelets. Patients with severe COVID-19 have elevated serum markers of neutrophil activation and NET formation that correlate with microthrombus formation, which is consistent with these observations [146].

In COVID-19 disease, a cytokine storm occurs where IL-1, IL-6, and TNF are the main inflammatory mediators released by infected cells [147]. They participate in coagulation by increasing tissue factor expression in epithelial cells [148] and releasing factor VIII and platelet-activating von Willebrand factor [149,150]. All of this has an impact on thrombin activation itself [151]. Aprotinin acts on all these processes contributing to thromboinflammation in COVID-19 (see Figure 4). It does this by inhibiting the classical complement activation pathway [128] and by inhibiting factors of the contact system that activate it, such as HMWK, plasmin, and coagulation factor XII [5]. Aprotinin is a drug that has a high capacity to inactivate the contact system by inhibiting kallikreins [27,28] and, in addition, other mediators that activate the contact system, such as proteases released from epithelial cells (see Table 1) or angiotensinase C released from damaged endothelium [40,41]. It also inhibits the lectin-mediated complement activation pathway by inhibiting MASP-2 and MASP-3 [47,48,49], as well as the alternative complement pathway [25,131] (Figure 3). Moreover, aprotinin inhibits the transmigration and activation processes of inflammatory cells such as neutrophils, which are the main players in causing acute inflammatory damage. These cells are one of its main targets. This is because aprotinin plays several roles: (i) it inhibits adhesion molecules such as P-selectin and CD11b by inhibiting Mac-1 expression [116,152] and also reduces the expression of intercellular adhesion molecule-1 (ICAM-1) on endothelial cells [116]; (ii) it inhibits the diapedesis of neutrophils by inhibiting the action of platelet-activating factor [38]; (iii) it inhibits matrix metalloproteases (MMPs) (i.e., matrix metalloprotease-2 (MMP-2) and MMP-9) [38] directly and indirectly, as these are activated by plasmin (for which aprotinin has a high inhibitory potency; Ki 90 pM). Metalloproteases form one of the main mechanisms for macrophage transmigration through the vascular wall [153,154]; (iv) it decreases IL-6 gene expression, as observed in rat myocardial cells after an ischaemia and reperfusion protocol [155], and it reduces the expression of IL-6 and TNFα in human tracheal epithelial cells [46], as well as IL-8 [38]; (v) it inhibits proteases such as neutrophil elastase and cathepsins, which also play a key role in neutrophil migration [25,29,32,156]; and (vi) it inhibits the secretion of phospholipase D and myeloperoxidase from neutrophils and thus decreases cell damage [157,158]. It is the inhibition of all these factors by aprotinin that prevents the processes of transmigration and inflammation. It is of note that the thromboinflammatory process is a cascade of activation events where some proteases activate others. Proteases such as thrombin or trypsin are potent activators of receptors activated by PAR-1 and PAR-2 proteases. These receptors, in turn, participate in a cascade of activation processes of other proteases, including interactions between the PAR receptors themselves [159]; membrane serine protease-1 [160]; other factors in the coagulation cascade [161], such as platelet activation factor [162]; and the production and release of renin [163]. This is the cause of the relationship between inflammation and thrombosis in SARS-CoV-2 infection and the reason that aprotinin is a potent drug to inhibit it by acting on this whole chain of events mediated by serine-proteases.

The third factor in Virchow’s triad is the impairment of vascular flow and permeability. In COVID-19 disease, mediators are released that affect vascular function. For example, angiotensin II, bradykinin, proteases (e.g., matrix metalloproteases and trypsin, among others), and inflammatory mediators such as cytokines themselves. Aprotinin has a high tropism for influencing the functionality of epithelia, including vascular endothelial cells and the lung epithelia themselves, impeding their permeability [152]. Angiotensin II and bradykinin are potent mediators of inflammation and are involved in oedema formation and vascular permeability. In addition, angiotensin II is pro-thrombotic, as it increases the production of plasminogen activator inhibitory peptide type I (PAI-1) in endothelial cells [164] and increases the expression of tissue factor that activates the extrinsic coagulation pathway [165]. This may contribute to the local microthrombus formation in alveolar capillaries that occurs in COVID-19 patients, as fibrin is not degraded by tissue plasminogen activator (tPA) and urokinase-type plasminogen activator (uPA) [166,167]. Moreover, bradykinin may also contribute to thrombosis by activating factor XII of the intrinsic coagulation pathway [168]. These peptide mediators are closely linked through the RAAS and KKS [74], and aprotinin has the capacity to reset the imbalance that occurs in these systems in COVID-19 disease [28]. In addition, aprotinin inhibits kallikreins and regulates coagulation by affecting factor XII, plasminogen, and PAR-1 thrombin receptors [44] (Figure 4). In fact, one of its clinical uses is to attenuate pulmonary vascular resistance in cardiopulmonary bypass surgery, which is triggered by inflammatory processes [169]. 

It is important to note that in COVID-19 disease, a state of hypercoagulation and impaired fibrinolysis occurs [170] which can be divided into three clinical stages depending on the severity of the patient: (I) elevated D-dimer; (II) elevated D-dimer, prothrombin time, and activated partial thromboplastin time, as well as thrombocytopenia; and (III) progression to a particular type of disseminated intravascular coagulation characteristic of COVID-19 [171]. These alterations in haemostasis occur due to an increase in coagulation that is explained by the relationship between complement activation and the contact system of innate immunity discussed above (Figure 3). This leads to the activation of the intrinsic pathway via kallikreins and factor XII (from the contact system), as well as the activation of prothrombin to thrombin by the action of multiple proteases [172]. In addition, thrombus formation is enhanced by the activation of the extrinsic pathway. High levels of angiotensin II or cytokines increase tissue factor expression in epithelial cells and their activation [148,165] (Figure 4). On the other hand, it is important to note that fibrinolysis is controlled by the humoral pathway of plasminogen activation to produce plasmin, in which factor XII, prekallikrein, and HMWKs are involved. The action of these factors shows a physiological antagonism on coagulation, by activating the intrinsic coagulation pathway while activating fibrinolysis through the action of plasmin [173,174]. This may explain the high levels of D-dimer production in patients with COVID-19 as a compensatory response to the prothrombotic state that occurs in the disease. In addition, such high levels of angiotensin II also increase the release of plasminogen activator inhibitor-1 (PAI-1), leading to a hypofibrinolytic state [123,164,167]. Finally, other coagulation control mechanisms are also altered. This includes (a) the urokinase system, which shows the increased plasma concentration of soluble urokinase receptors with the increased severity of infection [175,176] and decreased receptor expression on the epithelial cell membrane [177]; (b) the activated protein C and S systems which are decreased in patients with severe COVID-19 [178,179,180]; and (c) decreased antithrombin III activity [179,181,182]; while (d) α2-antiplasmin and α2-macroglobulin levels may be unchanged [167] or decreased [183,184] (see Figure 4). All these factors may contribute to local microthrombus formation in the alveolar capillaries. It is therefore understandable why in critical COVID-19 disease when initiating a process of sepsis, patients will experience a phenomenon of coagulopathy before sequentially developing a picture of disseminated intravascular coagulation [185,186]. These blood dyscrasia phenomena occurring in COVID-19 have similarities to thrombotic microangiopathy, which is caused by endothelial damage because of complement activation and a severe inflammatory process, where thrombotic processes can coexist with haemorrhagic ones [185,187]. By acting on the main mechanisms that cause these blood dyscrasias, aprotinin helps to control these imbalances.

### 3.4. Activity against the Symptomatic Processes of COVID-19

Many of the main symptoms of COVID-19 such as fever, cough or dry cough, pharyngitis, headache, myalgia, or dyspnoea [188], which occur with high frequency, are related to the inflammatory process associated with the activation of KKS proteases and complement innate immunity [189]. The ACE-2 enzyme, the anchor target of SARS-CoV-2, is involved in the clearance of bradykinin from the lung surface. As ACE-2 is internalised by viral entry, there is an increase in bradykinin in the lung, which is a potent irritant, proinflammatory, and vasodilator [84,88,89]. This may explain the dry and irritative cough in patients suffering from the infection. Dry cough is also associated with ACE-inhibiting antihypertensive drugs, as well as the occurrence of angioedema [190]. Aprotinin is a potent KKS inhibitor known to have antipyretic effects that are potentiated by non-steroidal anti-inflammatory drugs [191,192]. It also shows therapeutic efficacy in treating angioedema [193] and can reverse the dry, irritative cough caused by pharyngitis in COVID-19 [194]. The virus attaching to alveolar cells causes severe acute pneumonia, with an inflammatory response in which fibrinogen is released into the pulmonary alveoli, forming fibrin, and building up a hyaline membrane leading to pulmonary fibrosis, which is characteristic of SARS [195] and COVID-19 [196] infections. The greater this fibrin accumulation, the greater the risk of pulmonary microembolism [197,198]. Aprotinin is a potent fibrin inhibitor [5] and can thus prevent the harmful effects of pulmonary fibrin accumulation from occurring. In addition, SARS-CoV-2 induces an imbalance in the action of proteases and antiproteases in the infectious process [107,199,200]. The release of proteases by the respiratory epithelium is part of the innate immune response, to produce mucus mucin and collectins (pulmonary surfactant proteins) that support proper lung function and mucociliary clearance [201,202]. However, the excessive release of proteases, if not counterbalanced by the action of antiproteases, can exacerbate the inflammatory process, leading to chronic lung pathologies [203]. Aprotinin inhibits 80% of the proteases that exist in sputum and are secreted from oral, nasal, and pharyngeal epithelia in response to an infectious process. It therefore has positive effects on patients with chronic obstructive bronchitis by effectively dampening the pulmonary inflammatory process and mucus secretion [204]. 

Interestingly, epithelial sodium channels (ENaC) are also known to have a furin-like domain that is of vital importance for their activation. These domains are cleaved by serine proteases such as cathepsins, trypsin, matriptase, kallikreins, plasmin, prostasin, and urokinase [205,206]. Proper activation of these channels is also critical for airway surface fluid homeostasis, as well as cardiovascular and renal function [206]. It is well known that their poor regulation is associated with respiratory conditions such as cystic fibrosis [207]. Furthermore, in recent years, altered functionality has also been linked to immune cell activation, endothelial cell dysfunction, aggravation of inflammation involved in high salt-induced hypertension, pseudohypoaldosteronism, and tumour development [207]. Because of the exacerbation of serine-protease action that occurs in COVID-19 [107,199,200], the functionality of ENaC channels is altered [208,209,210,211]. This relates to many of the symptoms that occur in the disease involving mucus production, such as rhinorrhoea, nasal congestion, and expectoration). Respiratory distress is another example where ENaC would be implicated. Although this is closely related to the damage caused by the virus to type II pneumocytes, which affects pulmonary surfactant production [212], another cause is fluid accumulation affecting gas exchange and resulting in hypoxaemia. The formation of pulmonary oedema involves impairment of pulmonary fluid homeostasis, in which ENaC plays a major role [208]. This process involves the fibrinolytic system and the accumulation of fibrin in the lung, which not only causes altered channel function but also participates in inflammation and fibrin deposition itself, contributing to hypoxaemia and pulmonary fibrosis [209,210,211]. Furthermore, it is known that electrolyte disturbances occur in SARS-CoV-2 infection, most notably hypokalaemia, due to an imbalance in renal function through ENaC [213,214]. In the gut, its alteration has been suggested to contribute to the diarrhoea that occurs in COVID-19 [215] and to the vascular endothelial damage vasculopathy seen in SARS-CoV-2 infection [216]. Indeed, diuretic drugs that act on ENaC, such as amiloride, have been proposed to treat these channel-associated COVID-19 conditions [217].

Aprotinin inhibits the action of ENaC through its own inhibition of pulmonary proteases [218,219]. In addition, it blocks channel-associated Na+ currents [55] and has also been shown to regulate expression in the apical membrane of bronchial epithelial cells [220]. This contributes to normalising mucus viscosity and water balance in the airway and to restoring mucociliary clearance [205,221]. The same efficacy has been shown in conditions related to ENaC in other organs, such as the kidney [222]. 

## 4. The Clinical Efficacy of Aprotinin for Treating SARS-CoV-2 in Experimental Animals and in Humans

There are several families of viruses whose mechanism of infection is to induce a conformational change of a protein in their viral capsid by cleavage of a protease in the host cells to allow anchoring to its target. These mechanisms are common in the virus families Paramyxoviridae, Orthomyxoviridae, Retroviridae, Reoviridae, Rotaviridae, Herpesviridae, Flaviviridae, Filoviridae, Hepadnaviridae, Togaviridae, Poxviridae, and Coronaviridae [2,3,8]. As a broad-spectrum protease inhibitor, aprotinin has unique characteristics for use as an antiviral drug against these viruses.

The first studies that demonstrated the antiviral activity of aprotinin in experimental animals such as mice and chicken embryos were against influenza virus (of the Orthomyxoviridae family). Aprotinin was administered parenterally (intramuscularly) at a dose of 650,000 kallikrein inhibitory units (KIU) every 6 h for 6 days. A reduction in viral load was observed due to the prevention of viral haemagglutinin cleavage and the infective process. In addition, it prevented the pulmonary thromboembolic and haemorrhagic processes associated with the infection [223]. Subsequently, the results were also corroborated when administered via the intraperitoneal route at lower doses of 2000 KIU/day [8]. However, the idea of treating influenza virus infections with nasally administered protease inhibitors was first explored with α-aminocaproic acid and aprotinin in animals, as well as in children, where promising results on replication rate and cytopathic damage were obtained. This was the first time that clinical results of the use of these drugs in humans were published [7]. Subsequently, the same research group demonstrated the efficacy of aprotinin against viruses of the Paramyxoviridae family in infected mice, and they corroborated the results previously obtained when treating influenza virus. In these studies, aprotinin was administered by the inhalation route by aerosol formation using a collision nebuliser. The drug was able to cure fatal haemorrhagic bronchopneumonia and normalise body weight in rodents. These results suggested that low doses of protease inhibitors (6 µg/mouse/day, which is equivalent to an activity of approximately 38 KIU/mouse/day) administered by aerosol inhalation could be used to treat respiratory diseases caused by viruses. The significance of this work is that it was the first application of inhalational methods that allowed the administration of low doses of aprotinin into the lung, showing efficacy in animal models [9]. These results were corroborated by other research groups in lethal pneumonia caused by Sendai virus, a virus of the Paramyxoviridae family, where aprotinin also showed efficacy when administered via the inhalation route in mice [11]. Finally, seven variants of influenza A virus (H1N1, H3N2, H5N2, H5N2, H6N5, H9N2) and one of type B were studied in renal cell lines and mice. Resistance to oseltamivir existed within these strains. Mice were given 0.1 mL of aprotinin intravenously at a dose of 2 mg/kg one day after infection, twice daily, for five days. The drug was found to improve survival and normalise the weight of the mice after 14 days of follow-up. Furthermore, aprotinin was able to inhibit viral replication for all viral strains in the cell lines, including oseltamivir-resistant strains with a potency (EC50) of between 14 and 110 nM [224]. Aprotinin was shown to be non-resistant and had a broad spectrum of activity against influenza virus. An important milestone was the first human clinical trials to demonstrate the safety and clinical efficacy of inhalational aprotinin. The first trial was conducted on 85 patients (41 control and 44 aprotinin) diagnosed with influenza and parainfluenza. The drug was administered as an aerosol via the inhalation route, at a dose of 32,500 KIU/day (administered in three doses of ~10,800 KIU). The results showed the full recovery of ~82% of the patients after 6 days of follow-up. Symptoms of fever, headache, sore throat, hoarseness, and cough were evaluated. Only ~18% retained some symptoms of the disease in that period of time, although they showed improvement. All of those included in the treated group recovered. In contrast, only ~46% of the placebo group recovered, and ~12% showed no improvement at all. Aprotinin had no adverse effects [10]. This clinical trial allowed aprotinin to be marketed in Russia and Eastern European countries under the name Aerus® in aerosol form, being indicated as a locally acting respiratory antiviral in mild and moderate influenza disease [4] (see Figure 1B). After the outbreak of the COVID-19 pandemic, the immediate need arose to search for current antiviral drugs for therapeutic repositioning, to avoid excessive delay in their application and to avoid the costs involved in the discovery of a new drug. Because coronaviruses are viruses that use host proteases in their viral infection mechanism [2,3], aprotinin was well positioned to be used due to the previous knowledge of it. The first step to be taken in this direction was to demonstrate that aprotinin inhibits the viral infection of the novel SARS-CoV-2 coronavirus. Thus, in an in vitro study by Bojkova et al. (2020), aprotinin was shown to inhibit the replication of several SARS-CoV-2 strains (FFM1, FFM2, FFM6, and FFM7) in immortalised colorectal adenocarcinoma epithelial cell lines (caco-2), lung adenocarcinoma epithelial lines (calu-3), and in primary cultures of bronchial epithelial cells. They found that aprotinin had a potency in terms of 50% inhibitory concentration (CI50) ranging from 4 to 20.6 KIU/mL to inhibit the cytopathic damage caused by SARS-CoV-2 and its viral replication [18]. These results were corroborated at the same time in calu-3 cell lines, where aprotinin reduced viral load in cells at similar concentrations [19].

Ivashchenko et al. (2022) showed that aprotinin administered intraperitoneally for four days twice daily at a dose of ~650 KIU in mice infected with influenza virus, and ~1100 KIU in mice infected with SARS-CoV-2, in combination with remdesivir, monupiravir, favipiravir, nirmatrelvir, or AV5080, reduced the viral load in the lungs of the animals for both types of infection in all combinations. The combination of drugs increased the survival of infection and recovered the body weight of the animals [225]. The same research group conducted an interesting study in Syrian hamsters, where the aim was to investigate the prophylactic effect of aprotinin administered intranasally at a dose of 1000 KIU one hour before infection with SARS-CoV-2, and for three days, at which time the animals were killed. The treatment prevented infection in five animals and significantly reduced infection in three others compared to controls. Thus, the replication and spread of viral infection were prevented. In the same study, a prospective clinical trial was conducted during 3 months of observation, involving 32 health professionals working continuously at a high risk of infection.

Participants were administered aprotinin by spray twice daily (in the morning before entering the risk zone and in the evening after leaving) at a dose of 400 KIU twice daily (800 KIU/person/day). As supportive therapy, participants received Galavit^®^, an immunomodulatory agent (two 25 mg tablets twice daily, sublingually) and Kipferon^®^, a recombinant human interferon-α2 inducer (one suppository, twice daily). All study subjects were tested weekly for SARS-CoV-2 using the polymerase chain reaction method. At Week 6, all hospital workers were tested for specific IgG. The results showed that only 2 out of 30 workers (6.7%; 93.3% protection) were infected with SARS-CoV-2. Those infected had mild or no symptoms. The infection rate among hospital workers was 29%. No adverse events were reported [226]. This article is of great importance as it is the first to suggest the use of aprotinin in the prophylaxis of SARS-CoV-2 infection. Finally, the same research group conducted a clinical trial to assess the efficacy of aprotinin in patients with a moderate prognosis of COVID-19. Aprotinin was administered in combination with favipiravir or hydroxychloroquine. The route of administration was either intravenously at a dose of 1,000,000 KIU per day for 3 or 5 days, or by inhalation at a dose of 625 KIU, four times a day for 5 days. The combination of aprotinin with favipiravir was the most effective. The median recovery of clinical parameters (C-reactive protein, D-dimer, body temperature, improvement of clinical status, and hospital discharge) was between 3.5 and 5 days [227]. However, in a study of COVID-19 patients to assess the efficacy of aprotinin administered intravenously at a dose of 2,000,000 KIU four times daily, alone or in combination with anakinra (an IL-1 antagonist), to assess the inhibition of the thrombo-inflammatory response, there was not a satisfactory result [228]. 

Finally, a randomised phase III study in patients with moderate to severe COVID-19 called ATAC showed that aprotinin administered by nebulisation at a dose of 500 KIU shortened treatment time and reduced hospital stay by 5 days compared to the control group. These patients required less supplemental oxygen and had no adverse reactions to treatment [20]. In addition, it reduced viral load at five days of treatment, correlating with disease recovery [21]. 

Studies in experimental animals and human clinical trials suggest that aprotinin is a potential drug for the prophylaxis and treatment of SARS-CoV-2 viral infections and other respiratory viruses. The drug has several advantages compared to other antivirals used against SARS-CoV-2, such as a broad spectrum of activity, ability to be aerosolised, few drug interactions, low toxicity, and low cost. However, more clinical trials are required, particularly in early phases, to help to establish its safety and efficacy margins. In addition, it will be important to understand the possible mechanisms of resistance to aprotinin action developed by SARS-CoV-2 and its variants.

Because clinical trials evaluating the safety and efficacy of aprotinin as an antiviral were developed primarily in Russia, its use with this indication is approved in those countries that have agreements between their drug agencies and the respective Russian agencies. This is the case in Eastern European countries (for example, the Czech Republic) and, recently, China or India. This indication has been extended to infections by the SARS-CoV-2 virus.

## 5. Aprotinin Pharmacokinetics

### 5.1. Pharmacokinetics in Experimental Animals

In rats, when administered as an intravenous bolus, aprotinin forms high molecular weight aggregates by self-aggregation or by binding to other macromolecules or peptides. This may constitute a form of sustained release of aprotinin into plasma as it slowly disintegrates [229]. It is distributed in the extracellular space, which indicates that it crosses vascular barriers but remains distributed extracellularly (in the interstitium). Furthermore, 90 min after administration, it is mainly concentrated in the kidney (~50%), liver (~6%), and urine (~1%). The renal concentration is much higher than the plasma concentration, indicating a marked accumulation. Unlike in the kidney, it does not accumulate in the liver [230,231]. Its distribution and elimination follow bicompartmental pharmacokinetics. It is eliminated by renal excretion. Glomerular filtration is not dependent on the ionic charge of the molecule [232]. Thus, after being filtered by the glomeruli, aprotinin is actively reabsorbed in the proximal convoluted tubule and does not reach the pars recta [233,234]. Once it reaches the proximal convoluted tubule, aprotinin undergoes renal tubular reabsorption by glycoprotein 330/megalin (gp-330/megalin) and by the α2-macroglobulin receptor, also called LDL receptor-related protein type 1 [56]. Here, it undergoes an intense but slow accumulation process [230]. After 24 h, a significant percentage of the dose is already retained, mainly unmetabolised, in the renal tubules [235]. It is mainly stored within phagolysosomes at the apical border of proximal tubule cells [233,234]. Enzymatic inactivation occurs by breaking disulphide bridges in its catalytic region [229]. In addition, it can be metabolised in the proximal tubule either by luminal hydrolysis by brush border proteases followed by reabsorption of the resulting amino acids; or by luminal reabsorption by endocytosis followed by lysosomal degradation. Different isoforms of dipeptidyl peptidase are among the enzymes involved in its cleavage [36,236,237]. Although aprotinin is reabsorbed at the apical membrane of renal cells, it does not cross the basolateral membrane and therefore does not enter the systemic circulation [233,234]. Finally, it is detected in the basal portion of the collecting tubule cells colocalised with kallikrein. The presence of aprotinin in collecting tubule cells offers a partial explanation for its renal effects [5,235]. The clearance is 0.57 ± 0.02 mL/min/g kidney weight, corresponding to a glomerular filtration rate of 0.88 mL/min/g [233,234]. Only 0.8% of the dose administered and filtered in the glomerulus appears in the urine [232,233,234]. The amount of aprotinin in urine is less than 1% of the renal content.

### 5.2. Pharmacokinetics in Humans

Aprotinin has been approved for therapeutic use since 1959. Currently, aprotinin is marketed as Trasylol^®^ by Nordic Pharma S.A.U. Laboratories in the form of a solution for intravenous infusion (Figure 1A). In humans, aprotinin is administered in doses of 1–2 million KIU in a slow intravenous infusion of 30 min. After this, a continuous infusion of 250,000–500,000 KIU/h can be started until the end of the procedure without exceeding 7,000,000 KIU in total. After slow intravenous administration (30 min) and once in the blood, aprotinin is 80% bound to albumin and distributed in the extracellular space with a volume of distribution (Vd) of 26.5 L [238,239]. This indicates that, as in experimental animals, it passes through the blood vessels and is distributed extracellularly (interstitium). It is mainly concentrated in the liver and accumulates in the kidney [240,241]. Distribution and elimination follow bicompartmental pharmacokinetics. The elimination half-life (t1/2) in the alpha (fast) phase is 0.3 to 0.7 h, and in the beta (slow) phase is 5 to 10 h [239]. Plasma clearance is 7.6 ± 2.4% dose/hour in the fast phase and 0.4 ± 0.2% dose/hour in the slow phase [241]. Elimination is by renal excretion. Aprotinin appears not to affect blood pressure, glomerular filtration rate, total renal plasma flow, or renal excretory function [242]. A total of 24 h after intravenous administration, 42% of the dose is filtered mainly unmetabolised [238]. Once it reaches the proximal convoluted tubule, it undergoes renal tubular reabsorption via the glycoprotein gp-330/megalin and the α2-macroglobulin receptor. Drugs such as aminoglycosides or polymyxins can inhibit its renal absorption [56]. Only 5% of the dose administered and filtered in the glomerulus is eliminated as active via the kidney [238]. A proportion of aprotinin appears as metabolites (peptide fractions of the original molecule) [243]. Proteolytic cleavage occurs slowly by lysosomal enzymes in the kidney. Renal insufficiency causes a substantial decrease in aprotinin clearance, with a consequent increase in the elimination half-life and area under the aprotinin curve [238,244,245].

### 5.3. Pulmonary Administration of Aprotinin

#### 5.3.1. Administration Devices

Inhaled aprotinin can be administered through small, practical, and easy-to-use devices such as nebulisers, which were used in the ATAC clinical trial [20,21], where the drug was administered via an ultrasonic-type nebuliser using a vibrating piezoelectric mesh. This was performed following the recommendations of the Non-invasive Ventilation (NIV) Working Group of the Spanish Society of Emergency Medicine (SEMES), which recommends the use of vibrating mesh devices, preferably with a mouth pipette interface or with a one-way expiratory valve identical to the reservoir oxygenation system. In addition, it is recommended that a surgical mask be placed over the device to minimise particle dispersion. Vibrating mesh technology transforms the liquid drug into a fine vapour, with atomisation into small particles (2–5 µm) that reach the bronchial and alveolar levels [246] (see Figure 1C,D). Pressurised metered-dose inhalers can also be used, such as in the administration of the Aerus^®^ pressurised aprotinin aerosol. The device uses two types of interchangeable mouthpieces, one for nasal delivery and one for pulmonary delivery by inhalation [247] (see Figure 1B). Dry powder inhalers are also an option for delivering inhalational aprotinin, but this approach has disadvantages in that the drug has to be in the form of a fine powder and a high inspiratory flow rate is needed over 30–60 L/min, which may not be possible in the case of SARS-CoV-2 patients. 

#### 5.3.2. Dose by the Pulmonary Route

In the ATAC clinical trial, the dose of inhaled aprotinin was determined by considering multiple factors. Aprotinin is highly effective in preventing viral replication (measured as inhibition of viral N and M protein expression), cytopathic damage (measured as cell lysis and cell syncytia formation), and apoptosis (measured as inhibition of caspase 3 and 7 activation) caused by SARS-CoV-2. The drug showed an IC50 of between 4 and 20.6 KIU/mL to inhibit these viral actions. In addition, it has shown efficacy against SARS-CoV-1 (IC50 of 118 KIU/mL) [18]. We therefore know that 200–300 KIU will inhibit the entire cytopathic damage of SARS-CoV-2. Thus, to achieve these concentrations (KIU) in the respiratory tract, the technical considerations of the vibrating mesh nebuliser to be used must be considered. In the case of the ATAC clinical trial, the InnoSpire Go HH1342^®^ (Philips, Amsterdam, The Netherlands) and MicroAir U100^®^ (Omron, Kyoto, Japan), or similar, were used. Applying the Guyton equation [248], we found that the dose to be administered to the patient was 500 KIU (the dose to be put in the device to allow 200–300 KIU to reach the lungs). About the dosing regimen, it is necessary to consider the clinical experiences with this drug in other human clinical trials for this route of administration, where the pulmonary surfactant lining the respiratory epithelium is thought to be replaced every 1–2 h and may contribute to aprotinin clearance [10,247]. However, protein clearance in the pulmonary alveolus is known to be a slow process and may be impaired when pulmonary oedema is present [249,250]. Therefore, we decided that the dose and dosing schedule should be 500 KIU every 6 h, for a total dose of 2000 KIU/person/day [20,21]. A similar dose by the inhalation route in humans (1500–2000 KIU/day, in six inhalations throughout the day) was proposed by Zhirnov for the treatment of influenza virus infections [4]. A dose of 500 KIU of aprotinin via the inhalation route is about 2000 times lower than that usually administered intravenously. At these doses administered via the pulmonary route, the drug is not expected to access the systemic circulation in a sufficient quantity to have a systemic effect [249,250]. Aprotinin should be administered until the patient has a negative RT-PCR result or a negative antigen test and the symptoms of the inflammatory process have disappeared. Importantly, after the administration of 1,000,000 KIU by the intravenous route, therapeutic plasma concentrations of 147 ± 61 KIU/mL are achieved, inhibiting proteases at that level [239]. The inhalation route achieves concentrations of 200–300 KIU in the lung. These concentrations are therefore sufficient to inhibit most of the proteases in the lung epithelium (see Table 1).

Interestingly, aprotinin has been studied as an adjuvant to improve the bioavailability of pulmonary administered drugs of a peptide nature (i.e., insulin, calcitonin, leuprolide, recombinant granulocyte colony growth factor stimulants, thyroid-releasing hormone, or leptin). The rationale for their use is to prevent their metabolism by pulmonary proteases in the respiratory tract, and thus have a higher absorption capacity. The tests were performed on experimental animals such as rats [251,252,253] and rabbits [254]. The doses of aprotinin tested by the inhalation route ranged from 0.25 mg to 1 mg, which in enzyme activity values is equivalent to doses of approximately 1600 to 6300 KIU. This is much higher than 500 KIU, as in many of these cases they were administered via the endotracheal route [251,252]. The significance of these studies is that aprotinin was administered as an excipient.

#### 5.3.3. Pharmacokinetics of Aprotinin via the Pulmonary Route

Aprotinin is a naturally occurring protein that has a Kunitz-type domain. This means that it inhibits the activity of serine-protease enzymes. These enzymes are physiologically released from the lung glands [255,256]. Aprotinin is found in numerous bovine tissues and is also synthesised in mast cells. Bovine lungs contain large amounts. If this process of synthesis and release occurs naturally in the lungs, it is logical that there is a mechanism for the pulmonary clearance of these proteins. The nature of aprotinin is polybasic, with an ionic charge (isoelectric point pH 10.5) and molecular weight (6511 Da) that make it difficult for it to pass through the lung epithelium [5]. Considering that the dose of inhaled aprotinin is about 2000 times lower than that usually administered intravenously, it is not expected to enter the bloodstream in sufficient quantities to have a systemic effect. Considering that its Vd is 26.5 L, the plasma concentration would be extremely low upon entering the systemic circulation. As a result, if administered by inhalation, it is expected to act locally on the alveolar surface of the lung. Mechanisms that allow protein clearance at the alveolar level include mucociliary clearance, intra-alveolar degradation, and phagocytosis by alveolar or intravascular macrophages [257]. However, the most relevant processes are endocytosis through the alveolar epithelium, transcytosis, or paracellular diffusion across the epithelial barrier [250]. Part of the dose of aprotinin administered via the inhalation route will be eliminated by mucociliary clearance, which is an innate defence system against foreign agents [258,259]. In addition, protein transport across the respiratory epithelium is of great importance and has been extensively studied as a critical process for recovery from pulmonary oedema. Protein clearance is known to be mediated by different mechanisms depending on the protein concentration in the air spaces. At low concentrations (e.g., less than 0.5 g/100 mL albumin), the predominant mechanism is receptor-mediated endocytosis, and this follows kinetics that are saturable. In contrast, if concentrations are higher (greater than 0.5 g/100 mL albumin), absorption is via passive paracellular mechanisms, where the smaller the protein size, the easier it is to pass through [249,250]. It is important to note that a dose of 500 KIU is equivalent to approximately 0.076 mg of aprotinin. Therefore, these are very small amounts of protein in the lung which, under normal conditions, suggests that aprotinin will follow the mechanism of receptor-mediated endocytosis clearance. Among the protein transporters in the lung are known polymeric immunoglobulin receptors, which clear proteins such as albumin, including the receptor for α2-macroblobulin [249,260]. In particular, the α2-macroglobulin receptor has been described as able to take up aprotinin [56]. It is found in the bronchial and alveolar epithelium expressed on fibroblasts, dendritic cells, and alveolar and vascular macrophages [260]. Aprotinin uptake by the reticuloendothelial system therefore plays an important role in its elimination when administered by the pulmonary inhalation route [261,262]. Importantly, α2-macroglobulin receptors are also involved in the clearance of amyloid proteins, including the amyloid precursor proteins that characterise Alzheimer’s disease [263]. Because radiolabelled aprotinin binds to the fibrils of different types of amyloid [264], it has been used with high diagnostic sensitivity to detect amyloidosis by gamma imaging techniques [265,266]. This may indicate that aprotinin follows this clearance mechanism. Another endocytosis-mediated transporter that recognises aprotinin and is involved in its pulmonary clearance is gp-330/megalin [56]. It is one of the main receptors that mediate albumin endocytosis. It belongs to the low-density lipoprotein (LDL) receptor family, as does the β2-macroglobulin receptor. It is expressed in polarised epithelial cells such as type I and II pneumocytes, as well as in alveolar epithelial cells [260,267,268,269,270]. Type I and II pneumocytes are known to be important cells in protein metabolism in the lung and are one of the main barriers to protein passage at the systemic level [271]. Once aprotinin enters the cellular interior by receptor-mediated endocytosis, proteolytic cleavage will occur, which may be by slow degradation by lysosomal enzymes or through the action of enzymes such as the various dipeptidyl peptidase isoforms expressed in macrophages, bronchial, and alveolar epithelial cells. Dipeptidyl peptidases may furthermore be secreted into the alveolar fluid, where they may also contribute to the clearance of aprotinin before it is endocytosed [237,272]. Type I and II pneumocytes have anchored enzymes such as oligopeptidases (EP24.15) on their cell membranes, which are involved in the degradation of peptides such as bradykinin and angiotensin, as well as in the internalisation of their receptors [273], or neprilysin (EP24.11), which is a metallopeptidase involved in amyloid clearance [271,274]. These enzymes are recognised to have a high affinity for aprotinin [50,51]. Since these enzymes not only have the function of degrading peptides, but also of internalising proteins for degradation, and aprotinin has a high affinity for it, it is possible that it also uses these mechanisms in its degradation. A clinical problem in SARS-CoV-2 infected patients is respiratory distress syndrome, where there is a loss of integrity of the pulmonary microvasculature manifested by increased protein permeability [275]. This increased permeability can lead to sepsis when the reticuloendothelial system is unable to control the passage of pathogens [276]. Although in this situation the increased passage of aprotinin via transcytosis or paracellular diffusion mechanisms across the epithelial barrier is expected [249,250], aprotinin acts specifically on the inflammatory factors that produce it (e.g., inhibition of fibrinogen or the complement system), contributing to the amelioration of this pathological process. Therefore, as the pulmonary situation recovers, its passage into the systemic circulation will decrease. However, the concentration of aprotinin which may reach in plasma will always be lower than if administered parenterally.

## 6. Toxicity of Aprotinin by the Pulmonary Route of Administration

Aprotinin is a natural protein isolated from bovine lungs. Its use may therefore carry a risk of sensitisation in patients at a risk of hypersensitivity or anaphylactic reactions, especially with repeated use (Datasheet: Trasylol^®^ [aprotinin injection], 2003). Intravenous administration of Trasylol^®^ is indicated for prophylaxis to reduce blood loss and transfusion requirements in adult patients who are at a high risk of bleeding when undergoing isolated cardiopulmonary bypass surgery (i.e., coronary artery bypass surgery that is not combined with other cardiovascular surgery). It is also important to emphasise that other aprotinin-containing drugs, such as Artiss^®^ or Tisseel^®^, which come in the pharmaceutical form of tissue adhesive solutions, are widely used on the mucosal surfaces of surgical patients [277]. In this clinical practice, hypersensitivity and allergic reactions are extremely rare (there have been only 5 suspected adverse reactions after 1 million exposures to the fibrin–aprotinin adhesive) [278]. However, such adverse reactions call for caution with its use. These risks of hypersensitivity can be reduced by the production of recombinant humanised aprotinin using biotechnological techniques [31,279]. In addition, the risk of allergic/anaphylactic reactions is low in patients with no previous exposure but, in the case of re-exposure by the intravenous route, the incidence may be as high as 5%. For this reason, the European Medicines Agency (EMA) initially advised against measuring specific anti-aprotinin IgG antibody levels in patients to be administered aprotinin intravenously and advised against its use in those who have been exposed to the drug in the last 12 months and whose antibody levels are not known. A risk/benefit assessment was therefore recommended to be performed for all aprotinin-containing medicines before administration. All known safety data for the intravenous route (Trasylol^®^ [aprotinin injection], 2003) are listed in the data sheet for Trasylol^®^.

Prior to conducting clinical trials in humans, it had to be demonstrated that pulmonary administration did not produce toxic effects in the lungs. To this end, the cytotoxicity of aprotinin was first studied in vitro in immortalised A54 human lung cancer cell lines. No toxicity was observed in these cells by MTT assay, in contrast to other widely used drugs such as bacitracin, which showed toxicity at all concentrations tested [252]. Notably, aprotinin showed protective effects on the growth of metastases in some tumour types in vivo, preventing their spread to tissues, e.g., hepatoma 22, Lewis lung carcinoma [280], and breast carcinoma [281]. 

In addition, in vivo toxicological studies were performed after acute and chronic administration in two different species of experimental animals. In the acute toxicity studies, no alterations were found after examination of the functional status of the central nervous and cardiovascular systems, blood cell counts and blood biochemical analysis, and morphological and histological examination of all internal organs [282]. In addition, the effects of acute administration and after long periods of exposure by aerosol inhalation were studied in these experimental animals, and no allergic hypersensitisation and/or anaphylactic reactions were found to have occurred [283]. 

Safety assessments of clinical trials in which aprotinin was administered by inhalation have reported no adverse reactions. In patients with chronic obstructive pulmonary disease (COPD) and influenza virus-associated infections, inhaler administration has been well tolerated [204]. In addition, aprotinin has been used in the treatment of influenza and parainfluenza lung infections without notable adverse reactions. In clinical trials in 300 adult patients where it was administered by inhalation, no adverse reactions of an allergic or irritant nature were observed in any of the patients treated with it [10], and the same was true in children [7]. Similar observations have been made in the treatment of SARS-CoV-2 infection. In the clinical trial performed by Ivashchenko et al., which focused on aprotinin (IV or inhaled) in combination with Avifavir^®^ (favipiravir) or hydroxychloroquine, in hospitalised patients with moderate COVID-19-related pneumonia, no adverse events were recorded and all patients were discharged from hospital [226,227]. Similarly, in the ATAC study, no adverse reactions or side effects were reported in the aprotinin-treated group. One reason for this low incidence of adverse reactions is the low dose administered (2000 KIU/day) compared to that administered intravenously (3–7 × 106 KIU/day) [20,21]. Although no cardiovascular or renal adverse reactions are expected following inhalation administration of aprotinin, it is important to assess the possibility of local hypersensitivity/anaphylaxis reactions in patients, even if the doses to be administered by the pulmonary route are low. It is therefore important to perform a benefit/risk assessment of the patient before administering aprotinin via the pulmonary route, and to assess the group of patients who will benefit most from this treatment. Although it is advisable to measure aprotinin-specific IgG in previously exposed patients to prevent the possible occurrence of an allergic reaction, unfortunately, to our knowledge, there are no commercial tests available for this purpose. For this reason, if there is a risk of hypersensitivity, aprotinin can be combined with glucocorticoids to reduce the risk of this adverse reaction in these circumstances.

## 7. Conclusions

Aprotinin has a spectrum of activity that makes it a promising candidate for the treatment of COVID-19 disease, as well as other respiratory viruses that use host proteases in their mechanism of infection. In addition to its antiviral action, it has important anti-inflammatory, antithrombotic, and anti-symptomatic actions. Its inhalation route of administration allows high lung concentrations to be achieved with less likelihood of systemic adverse reactions. It is inexpensive, easy to administer, and has fewer drug interactions than other antiviral drugs used in the treatment of COVID-19 disease (e.g., favipiravir). However, its administration must be carried out with caution because it is a heterologous protein and due to the ever-present risk of allergic reactions. 

In its clinical development, for this new inhalation route of drug administration, pre-clinical knowledge of acute and chronic toxicity in experimental animals should be expanded to corroborate its safety when administered via this route. In addition, a new clinical study should be considered by means of a phase III trial to determine the dose–effect relationship in a few patients, as well as its safety in dose escalation. In this new study, it is important to correlate dose escalation with pharmacological efficacy, using predictive markers of clinical efficacy (e.g., viral load, measures of serum fibrinogen, bradykinin, concentration of secreted proinflammatory proteases in saliva) and toxicological efficacy (e.g., markers of immunogenicity through production of anti-aprotinin antibodies) that are reliable. Robust primary clinical benefit endpoints (e.g., sustained patient recovery time or an all-cause mortality analysis) should also be defined. Patients should be characterised virologically by viral strain analysis, viral load, and serological analysis, to study the drug response to different strains or mutations and their capacity for resistance. With the knowledge provided by these studies, and if the results are favourable, it could open the door to its commercialisation as a new antiviral drug. Its low price (~2 EUR/dose) will allow the treatment to reach countries with a lower economic capacity.

## Figures and Tables

**Figure 1 ijms-25-07209-f001:**
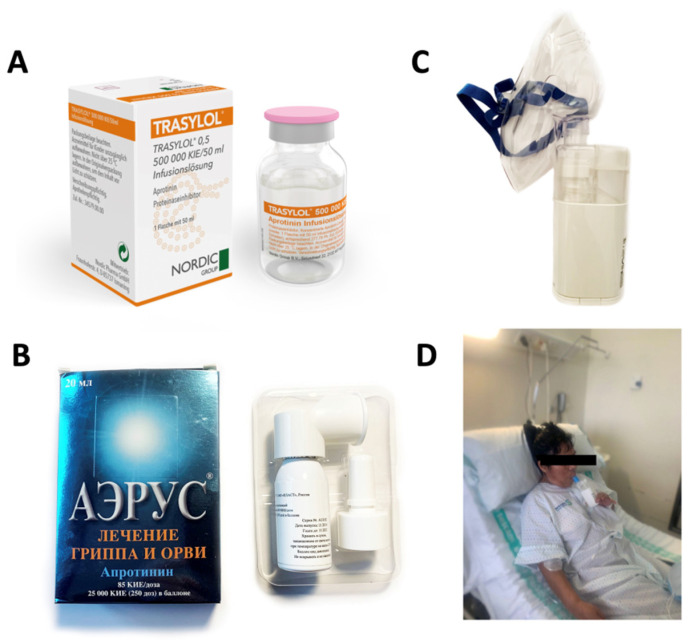
Dispensing forms of aprotinin for inhalational administration. (**A**) The aprotinin drug Trasylol^®^ for intravenous solution, marketed by Nordic Pharma SAU. In the clinical trial “Aprotinin Treatment Against COVID-19” (ATAC), this medicinal product was prepared for inhalational use by dilution in a 0.9% physiological saline sodium chloride solution to a dose of 500 KIU. (**B**) The aprotinin medicine Aerus^®^ is conditioned in a pressurised gas inhaler, delivering a dose of ±80 KIU per pulse. It has different types of mouthpieces to allow administration by inhalation or the nasal route. It is marketed in Russia and Eastern European countries by Nixdorf Preventive Health GmbH. (**C**) An ultrasonic nebuliser device using a vibrating piezoelectric mesh and a mask for delivery. (**D**) The first patient in the ATAC clinical trial treated with aprotinin using an ultrasonic nebuliser device.

**Figure 2 ijms-25-07209-f002:**
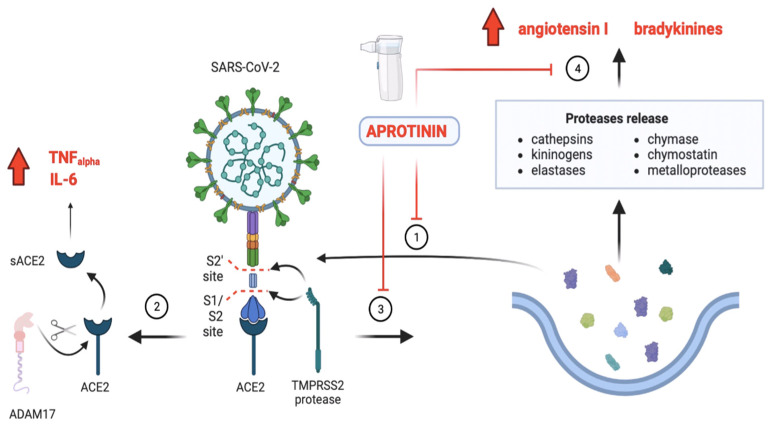
The main actions of aprotinin against COVID-19 disease. (1) Inhaled aprotinin inhibits 80% of the proteases released in the respiratory tract that are involved in the cleavage and activation of SARS-CoV-2 protein S for anchoring to the host cell entry target (antiviral mechanism of action), as well as in the bronchial hypersecretory and inflammatory responses (the main symptomatic mechanism of action). (2) Following the activation of the viral S protein by cleavage of the furin domains (S1/S2 and S2’ sites), SARS-CoV-2 anchors to angiotensin-converting enzyme type 2 (ACE2). Here, proteases such as ADAM17 upregulate ACE2 by downregulating its expression at the cell membrane and producing its soluble form (sACE2). This process contributes to the production of inflammatory mediators such as IL-6 and TNFα, which are the cause of the cytokine storm. Aprotinin prevents the release of these inflammatory mediators. (3) This process is aggravated by a vicious circle, where the very inflammation caused by the released proteases induces the release of further proteases. Factors such as oxidative stress prevent proper regulation by antiproteases such as α1-antitrypsin. (4) Hypersecretion of proteases such as cathepsins, kininogens, elastases, chymases, chymostatin, and metalloproteases increase the production of mediators such as bradykinin and angiotensin that cause inflammation, thromboinflammation, and the pulmonary symptoms of COVID-19 (e.g., dry cough or mucus). By preventing the action of these proteases, aprotinin will prevent all these processes.

**Figure 3 ijms-25-07209-f003:**
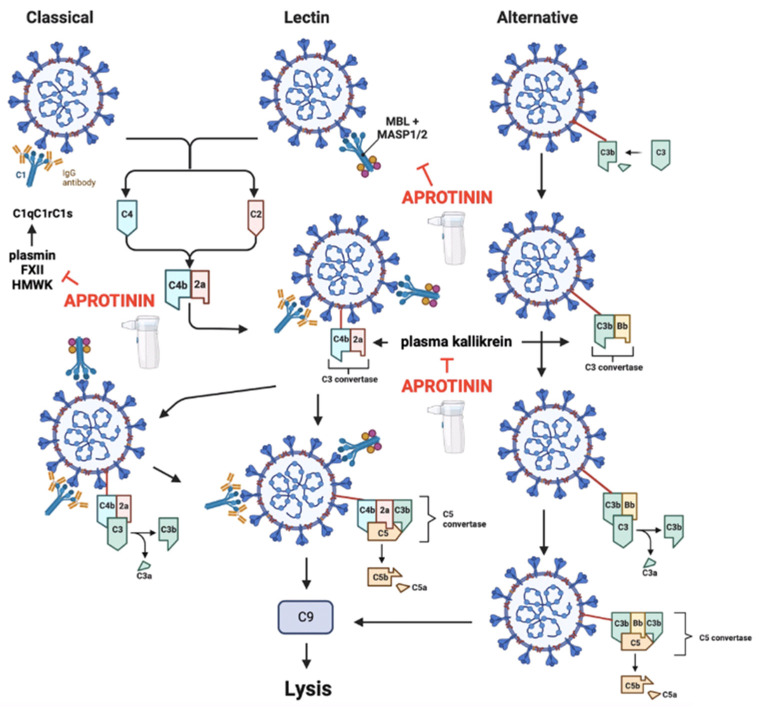
Actions of aprotinin on the complement pathway and the contact system of innate immunity. One of the main factors influencing the severity of COVID-19 disease is the hyperactivation of the innate immune system formed by the contact system (the kinin–kallikrein system, KKS) and the complement pathway. The process of protease hypersecretion caused by SARS-CoV-2 by the infecting epithelial cells releases, among many others, kininogens (e.g., high molecular weight kininogen, HMWK), plasmin, and coagulation factor XII (FXII), which are part of the contact system of innate immunity. They activate the classical complement pathway by binding to antigenic SARS-CoV-2 proteins and binding to the complement C1q complex, which activates this pathway. Aprotinin is a potent inhibitor of these cytogens. In addition, the complement lectin pathway is initiated by enzymes such as mannan-binding lectin serine proteases 1 and 2 (MASP-1/2) that cleave complement proteins C4 and C2 into fragments to form the C3 convertase complex. Aprotinin inhibits MASP-1/2 with high affinity as serine proteases (see Table 1) and is therefore also able to inhibit the complement lectin pathway. Finally, the C3 convertase complex can be formed either via the classical or lectin pathway of complement (C4b2a), or via the alternative pathway (C3bBb). This complex is a serine-protease that can be activated by kininogens such as kallikreins. Aprotinin is also a good inhibitor of these C3 convertase complexes.

**Figure 4 ijms-25-07209-f004:**
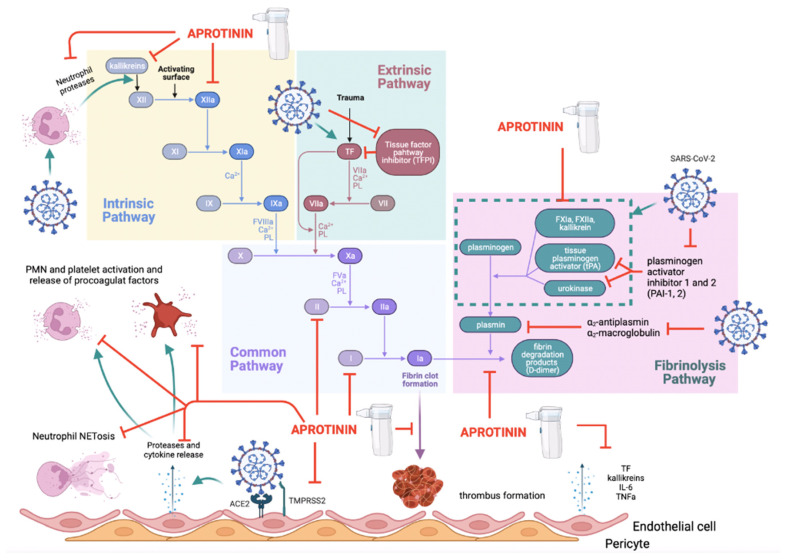
Actions of aprotinin on coagulation and “thromboinflammation”. SARS-CoV-2 in its invasive process releases proteases from inflammatory or epithelial cells, a factor that contributes to the infective process. This increases inflammation and, as these proteases are related to coagulation, also “thromboinflammation.” Aprotinin is a potent inhibitor of the contact system or KKS of the innate immune system. In addition, it inhibits the serine-proteases (e.g., kallikreins or elastases) which from kininogens sequentially initiate the cascade of events of the intrinsic coagulation pathway. However, SARS-CoV-2 is capable of releasing tissue factor (TF), which activates the extrinsic coagulation pathway and also inhibits the release of tissue factor inhibitor (TFPI). Both pathways, intrinsic and extrinsic, converge in the common coagulation pathway, where aprotinin inhibits both the formation of thrombin (Factor IIa) and fibrin (Factor Ia) for clot retraction through the latter. Coagulation is finely regulated by the fibrinolysis pathway. Paradoxically, SARS-CoV-2 also induces fibrinolysis through the release of kininogens (these activate the intrinsic pathway and fibrinolysis), tissue plasminogen activator (tPA), and urokinase that cause the conversion of plasminogen into plasmin, which it degrades fibrin into its degradation products (e.g., D-dimer). These mentioned factors are regulated by antiproteases such as α 2-antiplasmin, α2-macroglobulin, and plasminogen activator inhibitor types 1 and 2 (PAI-1, 2), which are all inhibited by SARS-CoV-2. Aprotinin regulates the fibrinolytic pathway both by acting on kininogens (factor XII or kallikreins), and by preventing the activation of protease-activated receptors (PARs), and directly inhibiting fibrinogen. For all these reasons, aprotinin prevents the processes of fibrosis and the formation of Neutrophil Extracellular Traps (NETosis).

**Table 1 ijms-25-07209-t001:** Targets of aprotinin and its role in COVID-19 disease.

	Targets	Provenance and Affinity	Participation in COVID-19	Bibliography
Proteases	Kallikreins (FXII—Hageman factor; HMWK—Fitzgerald factor)	They are of serum (Fletcher’s factor) and tissue origin, they are produced as a precursor (prekallikreins) 1–100 nM, pKi; plasma (pH 7.8–8.0) 0.8–1 nM, pKi; tissue 0.1 nM, pKi (urine, human, pH 8.0)	Angiotensin II production Bradykinin production Inflammation and fibrosis Symptomatic processes (cough, fever) Pulmonary oedema Thromboinflammation—clotting	Hoffmann et al., 1989 [27] Moreau et al., 2005 [28] Brinkmann et al., 1991 [29] Fritz and Wunderer, 1983 [5] Ivachtchenko et al., 2023 [26]
	Thrombin (FII)	It is synthesised in hepatocytes as a precursor (prothrombin) low affinity (61 μM, pKi)	Thromboinflammation—clotting PAR receptor agonist	Fritz and Wunderer, 1983 [5] Zhirnov et al., 2011 [4] Pintigny and Dachary-Prigent, 1992 [30]
	Plasmin	It is synthesised in hepatocytes as a precursor (plasminogen) 90 pM–4 nM, pKi (human and porcine, pH 7.8)	Activation by Viral S-Protein Cutting, Viral Entry Thromboinflammation—clotting Fibrosis NET Training	Fritz and Wunderer, 1983 [5] Ivachtchenko et al., 2023 [26] Sun et al., 2009 [31]
	Fibrinogen-fibrin (FI-FIa)	It is synthesised in the liver	Thromboinflammation—clotting Fibrosis NET Training	Fritz and Wunderer, 1983 [5]
	Tissue plasminogen activator (TPa)	Aprotinin has a Kunitz-like domain similar to tissue factor inhibitor peptide 8–27 μM, pKi	Thromboinflammation—clotting	Fritz and Wunderer, 1983 [5] Ivachtchenko et al., 2023 [26]
	Cathepsins	They are synthesised by epithelial, inflammatory cells. 10 nM, pKi (cathepsin G)	Angiotensin II production Activation by viral S-protein cut, and infectious process Endocytic entry of the virus Maturation of viral proteins in lysosomes Propagation and exocytosis of virions Heparanase release and cellular glycocalyx damage Inflammatory and symptomatic process Antigen presentation	Brinkmann et al., 1991 [29] Fahy et al., 1992 [32]
	Chymostatin	It is an anti-serine and cysteine-protease released from epithelial and inflammatory cells that potently inhibits cathepsin G or chymotrypsin 10 nM, pKi	Angiotensin II production Activation by viral S-protein cut, and infectious process Inflammatory and symptomatic process	Fritz and Wunderer, 1983 [5] Ivachtchenko et al., 2023 [26]
	Chymotrypsin	Glandular tissues 9 nM, pKi; (bovine pH 8.0)	Viral activation Inflammatory process	Fritz and Wunderer, 1983 [5] Ivachtchenko et al., 2023 [26]
	Trypsin	It is a serine protease released from epithelial and inflammatory cells 0.06 pM, pKi (bovine, pH 8.0)	Activation by viral S-protein cut, and infectious process PAR receptor agonist Inflammatory and symptomatic process Fibrosis Epithelial extracellular matrix damage	Fritz and Wunderer, 1983 [5] Ivachtchenko et al., 2023 [26] Brinkmann et al., 1991 [29]
	Chimases	Epithelial cells and myeloid cells ~30% inhibition to 15 μM	Produces angiotensin II by ACE-independent pathways	Lindberg et al., 1997 [33]
	Neutrophil elastase	Released by neutrophils, contained in their azurophilic granules 3.5 μM, pKi (pH 8.0). There are recombinant aprotinins that may have 0.4 μM pKi	Epithelial extracellular matrix damage Antiprotease inhibition Angiotensin II production Activation by viral S-protein cage and infectious process Oxidative stress Cytokine release NET formation, fibrosis Thromboinflammation	Fritz and Wunderer, 1983 [5] Ivachtchenko et al., 2023 [26] Brinkmann et al., 1991 [29]
	Tryptase TL2	16% inhibition to 10 μM	Viral infection and syncytia formation	Kido et al., 1990, [34] Brinkmann et al., 1997 [35] Fritz and Wunderer, 1983 [5] Ivachtchenko et al., 2023 [26]
	Dipeptidyl peptidase 3 and 4 (DPP-3 and 4)	Ubiquitous 11.7 µM pKi (DPP-3)	Viral entry as a target for the virus to anchor itself to the host cell Viral replication Inflammation Epithelial extracellular matrix damage Activation, proliferation, and transmigration of inflammatory cells	Engel et al., 2006 [36] Abramić and Agić, 2022 [37]
	Matrix metalloproteases (MMP2 and 9)	Zinc metalloproteases. Ubiquitous Inhibits 40% of its secretion at 100 μg/mL	Protein degradation, extracellular matrix and basement cell membrane Inflammation Coagulation	Kuyvanhoven et al., 2004 [38] Shu-Chen Chu et al., 2004 [39]
	Angiotensinase C	It is a polycarboxypeptidase of lysosomal origin. Structural homology with DPP-2	Angiotensin metabolism and production Inflammation Endothelial damage	Ripa and Gilli, 1968 [40] Dahlheim, 1972 [41]
	PAR-1 and 2	GPCRs activated by thrombin and trypsin. Intracellular signalling and intrinsic protease activity 50 KIU/mL (IC_50_) with PAR-1	Vascular remodelling Haematological alterations Inflammation Fibrosis Production and release of renin	Day et al., 2006 [42] Landis, 2007 [43] Khan et al., 2005 [44] Gomides et al., 2012 [45]
	Transmembrane serine protease 2 (TMPRSS2)	Epithelial cells	Viral entry as a target for the virus to anchor itself to the host cell Inflammation—cytokine storm	Yamaya et al., 2015 [46]
	Mannose-associated serine protease (MASP) 2 and 3	MASP-3, 8% inhibition, 10:1 ratio MASP-2, Ki 150 nM	Activation of the complement lecithin pathway Inflammation, immunity, clotting	Petersen et al., 2000 [47] Cortesio and Jiang 2006 [48] Keizer et al., 2015 [49]
	Oligopeptidases (EP24.15)	Cytosolic, associated with membranes and secreted to the outside of the cellProtease that degrades peptides such as bradykinin and angiotensin	Angiotensin production Inflammation, modulation of the immune system Symptoms of COVID-19	Vanneste et al., 1990 [50] Aoyagi et al., 1990 [51]
	Not too much (EP24.11)	Metallopeptides. Killers	Angiotensin metabolism Inflammation, immune response	Vanneste et al., 1990 [50] Aoyagi et al., 1990 [51]
Other targets	Nitric oxide synthase (iNOS)	Ubiquitous ~IC50 600 KIU/mL	Inflammatory process Inhibition of antiproteases	Hill et al., 1997 [52] Hill and Robbins, 1997 [53] Bruda et al., 1998 [54]
	ENaC	Epithelial cells 80% inhibition to 1 μM	Symptoms of COVID-19 (e.g., nasal congestion or diarrhoea)	Adebamiro et al., 2005 [55]
	α2-Macroblobulin receptor	Bronchial and alveolar epithelial fibroblasts, dendritic cells, and alveolar and vascular macrophages	Clearance of proteins in the lung, including aprotinin	Moestrup et al., 1995 [56]
	GP-330/Megaline receiver	Type I and II pneumocytes and alveolar epithelial cells IC50 4 μM	Clearance of proteins in the lung, including aprotinin	Moestrup et al., 1995 [56]

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
