# Peer review of "Aprotinin (II): Inhalational Administration for the Treatment of COVID-19 and Other Viral Conditions"

_ijms, 2024, doi:10.3390/ijms25137209_

Round 1

Reviewer 1 Report

Comments and Suggestions for Authors

The manuscript wrote well on Aprotinin (II) and COVID-19. I have minor few comments.

1. As the author said that Aprotinin II inhalation administration is good candidate and low cost why it dose not use it over the world?

2. Please describe the disadvantages of Aprotinin II againist COVID-19.

3. Aprotinin is only purpose for treatment, not for prevention ?

4. Aprotinin II inhibit SARS-CoV-2 in entry or replication step?

5. Which countries widely used for Aprotinin for COVID-19 treatment?

6. Except COVID 19 infection, Aprotinin II and other viral infection mechanism should be discussed in the manuscript.

Author Response

These are the changes suggested by reviewer 1:

  1. As the author said that Aprotinin II inhalation administration is good candidate and low cost why it dose not use it over the world?

As explained in the conclusions, the use of aprotinin as an antiviral via the inhalation route requires approval by the drug agencies of different countries (for example, FDA or EMA). Although a preliminary phase III study has been carried out (Redondo-Calvo et al., 2022a,b; quotes 20 and 21 in the manuscript), it is necessary to carry out more pre-clinical studies (mainly of acute and chronic toxicity in animals), and clinical trials in humans (dose escalation and phase III) as indicated in the conclusions from line 926 to 942.

  1. Please describe the disadvantages of Aprotinin II againist COVID-19.

The disadvantages of aprotinin are already discussed in the manuscript. Like any antiviral drug, resistance can occur. This is already reflected in line 633 to 635. The resistance mechanisms for SARS-CoV-2 have not yet been described. Although it can be speculated that, as happens for the influenza virus, because aprotinin has the capacity to inhibit multiple proteases, it is more difficult for the virus to evade its pharmacological action and, therefore, resistance is lower. On the other hand, more studies are necessary to better define both its safety and effectiveness. Although the drug appears safe, the possibility of hypersensitivity reactions in some patients should not be ruled out (especially if it were used in large populations of people). This is reflected in lines 905 to 915.

  1. Aprotinin is only purpose for treatment, not for prevention?

Your comment is appropriate. Although in the manuscript we talked about the study by Ivashchenko et al., 2020 (quote 220) on prophylaxis, we had not clearly highlighted this prophylactic use. Therefore, we have introduced the following sentences in the manuscript:

1) “This article is of great importance as it is the first to suggest the use of aprotinin in the prophylaxis of SARS-CoV-2 infection.” (lines 616 to 618)

2) “Studies in experimental animals and clinical trials in humans suggest that aprotinin is a potential drug for the prophylaxis and treatment of SARS-CoV-2 viral infections and other respiratory viruses (lines 635 to 637).

  1. Aprotinin II inhibit SARS-CoV-2 in entry or replication step?

The answer to your question is both. To clarify which steps in the viral infection process aprotinin inhibits we have inserted the following sentence in lines 168 to 174 of the manuscript. “In summary, due to the ability to inhibit multiple proteases (for example cathepsins), aprotinin has antiviral actions by preventing the attachment of the virus to the target protein (by preventing activation of viral protein S); penetration (prevent endocytosis or syncytium formation); the replication, maturation and trafficking of viral proteins to lysosomes; the assembly of virions and their release (for example, by inhibiting heparanase or angiotensinase C). The proteases it inhibits and their involvement in the viral infection mechanism are summarized in Table 1

  1. Which countries widely used for Aprotinin for COVID-19 treatment?

Aprotinin is used for the prevention and treatment of COVID-19 infections in countries such as Russia and Eastern European countries such as the Czech Republic. Recently, its use is being introduced in China and India.

Because drug agencies typically require that studies affecting safety and effectiveness (clinical trials) be done in their respective countries, and the drug was primarily developed in Russia, countries where this use is permitted are in those that have agreements with the respective agencies (in this case that of Russia). Unfortunately for this drug, this is not the case in Europe or the United States.

We have introduced the following sentence in the manuscript: “Because clinical trials evaluating the safety and efficacy of aprotinin as an antiviral were developed primarily in Russia, its use with this indication is approved in those countries that have agreements between their drug agencies and the respective Russian agencies. This is the case in Eastern European countries (for example, the Czech Republic) and, recently, China or India. This indication has been extended to infections by the SARS-CoV-2 virus.” (lines 643 to 648).

  1. Except COVID 19 infection, Aprotinin II and other viral infection mechanism should be discussed in the manuscript.

As the discussion of these mechanisms is very extensive, we have divided the manuscript into two parts, the first titled “Aprotinin (I): Understanding the Role of Host Proteases in COVID-19 and the Importance of Pharmacologically Regulating Their Function” where we discuss these issues deeply, and the second titled: “Aprotinin (II). “Inhalational administration for the treatment of COVID-19 and other viral conditions.” The Aprotinin (II) manuscript refers to this second part, which you have evaluated. I recommend reading the first part.

All changes are marked in yellow in the manuscript

Thank you.

Reviewer 2 Report

Comments and Suggestions for Authors

The review by Padin et al. described the effect of inhalational administration of Aprotinin (II) for the treatment of COVID-19 and other viral conditions.

The review is interesting and well organized and written. The English used is generally good. There are only some minor revisions I would like to suggest.

Line 149: There is a typo: ORF à Open Reading Frame

I suggest improving the quality of all figures, in particular Figure 1: the photos are very blurry.

I also suggest to re-arrange Table 1 as a three-line table: in the present form is not very clear; please specify which statement corresponds to the one in the following column.

Author Response

Comments 1: Line 149: There is a typo: ORFa Open Frame

Answer 1: the acronym has been changed 

Comments 2: I suggest improving the quality of all figures, in particular Figure 1: the photos are very blurry

Answer 2: 

All four figures have been changed to a higher quality.

The table has been modified as proposed

All changes are marked in yellow
